# Hydrogen Compression Materials with Output Hydrogen Pressure in a Wide Range of Pressures Using a Low-Potential Heat-Transfer Agent

**Xu Zhang [1], Yu-Yuan Zhao [1], Bao-Quan Li [1], Mikhail Prokhorenkov [2], Elshad Movlaev [2], Jin Xu [1], Wei Xiong [1], Hui-Zhong Yan [1],\* and Sergey Mitrokhin [2],\***

[1] State Key Laboratory of Baiyunobo Rare Earth Resource Researches and Comprehensive Utilization, Baotou Research Institute of Rare Earths, Baotou 014030, China; lzhangxu@126.com (X.Z.); zhaoyuyuan1028@126.com (Y.-Y.Z.); lbqxplbq@126.com (B.-Q.L.); xujinaou123@126.com (J.X.)

[2] Chemistry Department, Lomonosov Moscow State University, Moscow 119991, Russia; mikl1995@yandex.ru (M.P.); movlaev@hydride.chem.msu.ru (E.M.)

\* Correspondence: yhzmail@126.com (H.-Z.Y.); mitrokhin@hydride.chem.msu.ru (S.M.)

**Abstract:** In order to meet the demand of metal hydride–hydrogen compressors (MHHC) and their hydrogen compression materials for high-pressure hydrogen filling in a hydrogen energy field, four kinds of hydrogen storage alloys with low-grade heat source (<373 K) heating outputs and different hydrogen pressures (up to 80 MPa) were developed as hydrogen compression materials. The preliminary compositions of the hydrogen storage alloys were determined by using a statistical model and research experience. The rare earth series $AB_5$ and Ti/Zr base $AB_2$ hydrogen storage alloys were prepared using a high-temperature melting method. The composition, structure, and hydrogenation/dehydrogenation plateau characteristics of the alloys were tested by an inductively coupled plasma mass spectrometer (ICP-MAS), X-ray diffractometer (XRD), and pressure–composition isothermal (PCT) tester. The median output pressures of the four-stage hydrogen storage alloys at 363 K were 8.90 MPa, 25.04 MPa, 42.97 MPa, and 84.73 MPa, respectively, which met the requirements of the 20 MPa, 35 MPa, and 70 MPa high-pressure hydrogen injections for the MHHCs. In fact, due to the tilted pressure plateau of the PCT curve, the synergy between the adjacent two alloys still needed to be adjusted.

**Keywords:** hydrogen compression materials; hydrogen storage alloy; metal hydride hydrogen compressors; hydrogen storage and supply; kinetics; thermodynamics; crystal structures





## 1. Introduction

Hydrogen energy has the advantages of having abundant resources, convenient storage, a wide application range, being clean and low-carbon, and a capacity for interconnection and collaboration [1–5]. $H_2$-$O_2$ fuel cells, as a model of hydrogen energy utilization, have entered the market introduction stage in the fields of transportation, energy storage, distributed power supply, and so on [6–9]. In particular, the power of fuel cell vehicle engines has been greatly improved, and they can run more than 700 km with a 70 MPa hydrogen storage tank [10]. Therefore, terminal hydrogenation stations will become a kind of popular public facility, but the high construction cost of these hydrogenation stations is the same dilemma faced by the global hydrogen energy industry. The use of a metal hydride–hydrogen compressor (MHHC) could reduce the current high input of these hydrogenation stations [1,5,11].

Advanced hydrogen storage and supply systems can use low-grade heat sources (such as industrial waste heat at T < 573 K) or solar heat to heat MH to dehydrogenate and convert $H_2$ into high-pressure and high-purity $H_2$. The main advantages of this are that low-grade heat sources are used instead of electricity, and the operation is simple,

with no moving parts, a compact structure, safety, and reliability. It is a better choice than conventional (mechanical) and newly developed (electrochemical or ionic liquid piston) hydrogen compression methods. Since the concept of MHHCs was proposed in the United States Patent (US3516263) in 1970, the research and development work, from principle to application, has become more and more in-depth. MHHCs have been used in aerospace, hydrogen isotope processing, water pump/actuator power, and other specific applications. Norway has built a 70 MPa demonstration hydrogenation station using MHHCs [1,12–14].

The main components of MHHCs are hydrogen storage alloy materials that can form MH [1,5,14–17], such as rare earth $AB_5$ intermetallic compounds [18–23], titanium/zirconium base AB and $AB_2$ intermetallic compounds [12,24–38], and vanadium base solid solutions [39]. The hydrogen storage alloy is accompanied by electron transfer and a change in hydrogen pressure and heat while reversibly absorbing/releasing hydrogen. At the same time, its equilibrium pressure changes exponentially with a change in temperature. A variety of alloys derived from the above alloy systems offer the possibility of hydrogen pressure outputs over a wide pressure range, using water as a heat transfer medium (T < 373 K). Recently, hydrogen compression materials with a pressure range of 3.2–85 MPa have been developed in China using low-grade heat sources, which can be used to fill high-pressure hydrogen storage devices for hydrogen energy applications [5]. The advanced hydrogen compression materials developed in low-pressure ranges (<25 MPa) are $AB_5$ type rare earth hydrogen storage alloys, and for a high-pressure (>25 MPa) hydrogen output, the preferred hydrogen compression materials are $AB_2$ type titanium base hydrogen storage alloys [5]. In this paper, considering the hydrogenation requirements of a long tube trailer with 20 MPa hydrogen and an on-board hydrogen tank with 35 MPa and 70 MPa hydrogen, hydrogen storage alloy materials of a 1–4 stage hydrogen compression for MHHCs were designed and prepared, with the highest output hydrogen pressure exceeding 80 MPa. Compared with the existing advanced hydrogen compression materials, the $AB_2$ titanium base hydrogen storage alloy with a more adjustable pressure range was used in the low-pressure range, and the plateau characteristics of all the stages of the hydrogen storage alloy were further optimized to improve the application performance of the MHHCs.

During the development of MHHCs at home and abroad, the research work on hydrogen storage alloys has mainly focused on the simulation, testing, and evaluation of the main properties of the materials. In fact, the composition, preparation technology, and technological conditions of these materials are very important for improving their application efficiency. In this paper, the basic composition of the hydrogen storage alloy used in the MHHCs was designed by the Moscow State University of Russia (MSU), according to its theoretical research results. The China Baotou Rare Earth Research Institute (BRIRE) further optimized the composition of the alloy through experimental verification and studied the preparation technology and process conditions. The industrialization technology and application products of 1–4 stage hydrogen storage alloy materials ($AB_n$) have been developed.

## 2. Results and Discussion

The preliminary compositions of the 1–4 stage hydrogen storage alloys were determined by using a statistical model developed in MSU, which is described in the literature [40]. Based on the existing research results and experimental work, BRIRE developed the alloy product composition and an ICP analysis confirmed that it was basically consistent with the design composition. The first alloy consisted of two rare earth series $AB_5$ types, the second and third alloys were Laves phase $TiCr_2$ types, and the fourth alloy was a Laves phase $TiFe_2$ type. Table 1 lists the design compositions and ICP analysis results of the first-stage rare earth hydrogen storage alloys. Due to the high vapor pressure of Ca elements, the content of the Ca elements in the composition of the First-2 alloy differs greatly from the design value and causes the fluctuation of the other components. Fortunately, the First-2 alloy products with the ICP results showed the expected properties, so no further alloy products were prepared in accordance with the design composition. Table 2 lists

the design compositions of the 2–4 stage hydrogen storage alloys and the ICP analysis results of the products. Table 3 lists the alloy numbers, compositions, test temperatures (T), maximum hydrogen storage capacities ($C_{max}$), and their corresponding hydrogenation/dehydrogenation pressures ($P_a$, $P_d$), hydrogenation/dehydrogenation plateau slope factors ($S_f$), hysteresis coefficients ($H_f$), reaction enthalpies ($\Delta H$), and entropies ($\Delta S$).

**Table 1.** Design composition and ICP analysis results of rare earth hydrogen storage alloy.

| Alloys | Compositions | La | Ce | Ca | Y | Ni |
|---|---|---|---|---|---|---|
| First-1 | Design | 19.84 | 5.00 | / | 5.29 | 69.86 |
| | ICP | 19.20 | 5.04 | / | 5.02 | 70.74 |
| First-2 | Design | 16.93 | 6.83 | 1.47 | 3.25 | 71.53 |
| | ICP | 16.11 | 6.32 | 0.80 | 2.71 | 74.06 |

**Table 2.** Design composition and ICP analysis results of 2–4 stage hydrogen storage alloys.

| Alloys | Compositions | Ti | Zr | Cr | Mn | Cu | V | Fe |
|---|---|---|---|---|---|---|---|---|
| Second | Design | 30.89 | / | 36.91 | 24.82 | 4.10 | 3.29 | / |
| | ICP | 29.90 | / | 37.44 | 25.72 | 4.22 | 2.72 | / |
| Third | Design | 31.09 | / | 47.27 | 7.14 | / | / | 14.51 |
| | ICP | 30.42 | / | 47.50 | 7.45 | / | / | 14.63 |
| Fourth | Design | 22.96 | 10.94 | / | / | / | 9.16 | 56.93 |
| | ICP | 23.03 | 10.45 | / | / | / | 9.96 | 56.56 |

**Table 3.** The main characteristics of 1–4 stage hydrogen storage alloys.

| Alloys | Compositions | T (K) | $C_{max}$ (H/f.u.) | $P_a$ (MPa) | $P_d$ (MPa) | $S_f$ | $H_f$ | $\Delta H$ (kJ·mol$^{-1}$ H$_2$) | $\Delta S$ (J·mol$^{-1}$·K$^{-1}$ H$_2$) |
|---|---|---|---|---|---|---|---|---|---|
| First-1 | La$_{0.6}$Ce$_{0.15}$ Y$_{0.25}$Ni$_{5.0}$ | 303 | 6.25 | 2.52 | 1.72 | 0.40 | 0.38 | −21.74 [a] | −98.69 [a] |
| | | 323 | 6.23 | 4.44 | 3.21 | 0.40 | 0.32 | −25.02 [d] | −106.25 [d] |
| | | 363 | 6.24 | 10.55 | 8.90 | 0.42 | 0.17 | | |
| First-2 | La$_{0.5}$Ce$_{0.2}$Y$_{0.15}$Ca$_{0.15}$Ni$_{5.0}$ | 303 | 6.60 | 2.05 | 1.44 | 0.23 | 0.35 | −22.59 [a] | −99.81 [a] |
| | | 323 | 6.51 | 3.73 | 2.71 | 0.21 | 0.32 | −24.34 [d] | −102.63 [d] |
| | | 363 | 6.09 | 9.08 | 7.15 | 0.25 | 0.24 | | |
| Second | TiCr$_{1.1}$Mn$_{0.7}$ V$_{0.1}$Cu$_{0.1}$ | 298 | 2.55 | 5.90 | 4.72 | 1.46 | 0.22 | | |
| | | 323 | 2.49 | 11.93 | 11.52 | 1.05 | 0.03 | −20.28 [a] | −102.14 [a] |
| | | 353 | 2.40 | 21.13 | 19.06 | 1.22 | 0.10 | −22.19 [d] | −106.78 [d] |
| | | 363 | / | 26.13 [c] | 25.04 [c] | / | / | | |
| Third | TiCr$_{1.4}$Mn$_{0.2}$ Fe$_{0.4}$ | 293 | 2.65 | 16.07 | 15.86 | 0.74 | 0.01 | | |
| | | 323 | 2.58 | 29.64 | 27.06 | 0.94 | 0.09 | −14.51 [a] | −91.89 [a] |
| | | 353 | 2.59 | 44.05 | 37.38 | 1.59 | 0.16 | −12.337 [d] | −84.37 [d] |
| | | 298 | / | 17.85 | 17.60 | / | / | | |
| | | 363 | / | 51.59 [c] | 42.97 [c] | / | / | | |
| Fourth | Ti$_{0.8}$Zr$_{0.2}$Fe$_{1.7}$V$_{0.3}$ | 293 | 2.90 | 35.0 | 31.50 | 1.24 | 0.10 | | |
| | | 323 | 2.63 | 61.50 | 48.00 | 1.38 | 0.24 | −13.11 [a] | −93.61 [a] |
| | | 353 | 2.58 | 87.13 | 77.36 | 1.33 | 0.12 | −12.75 [d] | −91.18 [d] |
| | | 298 | / | 39.05 | 33.70 | / | / | | |
| | | 363 | / | 100.75 [c] | 84.73 [c] | / | / | | |

[a]: The value was calculated by absorption plateau. [c]: Linear extrapolation value from the Van't Hoff diagram in Section 2.3. [d]: The value was calculated by desorption plateau.

## 2.1. Material Composition Design and Preparation

The application of MHHCs requires the consideration of two important technical indexes: the cycle yield and the compression ratio. The cycle yield is related to the reversible hydrogen storage capacity of hydrogen compression materials. The compression ratio is related to the hydrogenation enthalpy or plateau pressure and plateau hysteresis coefficient of hydrogen compression materials [1,5,41]. In addition, the activation properties, kinetic properties, plateau pressure coordination, flatness of the pressure plateau, and stability of the hydrogen absorption/discharge cycle need to be considered. In order to meet the MHHCs' requirements for hydrogen compression materials, the composition and preparation of these hydrogen compression materials are particularly important.

Reversible hydrogen storage capacity refers to the amount of hydrogen storage that can be released by hydrogen compression materials at a certain temperature, which is mainly

related to the type of materials and the type and content of the component elements [17,38]. In general, at the same temperature, the composition of a material with a higher proportion of hydrogen-absorbing elements has a higher hydrogen storage capacity. Increasing this reversible hydrogen storage capacity is beneficial to improving the MHHC cycle yield.

The enthalpy of a hydrogenation reaction is inversely related to the plateau pressure of the material [5]. Generally speaking, the larger the cell volume of the constituent phase of the hydrogen compression material, the lower the plateau pressure and the higher the enthalpy of the hydrogenation reaction. An increase in the absolute enthalpy is beneficial for improving the compression ratio of MHHCs, but a material with a high enthalpy value has a low pressure plateau and cannot achieve the purpose of a high-pressure hydrogen output.

Plateau hysteresis reflects the difference in the degrees of hydrogenation and dehydrogenation of hydrogen compression materials, which is caused by the stress of hydride formation [42]. The hysteresis factor ($H_f$), as shown in Equation (1) [32], is utilized in describing this plateau hysteresis.

$$H_f = \ln P_a / P_d \qquad (1)$$

The larger the cell size of the hydrogen compression material, the smaller the hysteresis of the plateau. Plateau hysteresis reduces the compression ratio of MHHCs, and at the same time, damages the cyclic stability of the material's hydrogenation/dehydrogenation [33,43].

The surface of the hydrogen compression materials needs to be in a good active state before the normal hydrogenation/dehydrogenation reaction, that is, the oxide layer that is formed on the surface by active elements such as La, Ce, Ti, and Zr in the material's composition must be removed. Doping some rare earth elements or rare earth mixtures in the material's composition can improve the activation properties of these hydrogen storage materials [44].

The kinetic properties of hydrogen compression materials are very important for improving the working efficiency of MHHCs. The contents of some of the metal elements in the material's composition, such as Ni, Cr, and Al, etc., and the state of its surface elements are related to the intrinsic kinetic properties of the materials [45].

The synergy of the plateau pressure between the two adjacent stages, i.e., the dehydrogenation pressure of the former-stage material at a high temperature is higher than the hydrogenation pressure of the latter-stage material at a low temperature, ensures the complete hydrogenation/dehydrogenation of each stage material.

The flatness of the material's pressure plateau is related to the interstitial site of the different volumes caused by the fluctuation of the material's composition [46,47], which mainly affects the synergy of the above plateau pressure and the kinetic performance. The slope of the pressure plateaus is usually characterized by slope factor $S_f$, which is described by Equation (2) [32]:

$$S_f = d(\ln P) / d(H \ wt\%) \qquad (2)$$

where $H$ is the mass hydrogen storage density of the alloys.

Resulting from the existence of $S_f$, the alloys absorb/desorb hydrogen incompletely at the midpoint of the plateau pressure.

The hydrogenation/dehydrogenation cycle stability of hydrogen compression materials during their application is related to the service life of MHHCs. The main factors leading to the cyclic stability of the materials are poisoning, disproportionation, and amorphous and lattice defects. Impurities such as CO, $H_2O$, and $O_2$ in hydrogen make the material toxic and its performance deteriorate. Water and oxygen can significantly poison Ti-based $AB_2$/AB-type alloys [48], while $AB_5$-type alloys are easily poisoned by CO at ppm amounts [49]. In comparison to $AB_5$-type alloys, the cycle life of $AB_2$-type alloys tends to be longer, which may be ascribed to the fewer defects they produce in the process of hydrogen absorption and desorption [50].

The development of all the stages of hydrogen compression materials also needs to consider the characteristics of an MHHC system operation. The hydrogen source used by

MHHCs should first adopt hydrogen produced by the electrolysis of water using renewable energy sources. At a normal temperature (293–303 K), the hydrogen output pressure is usually greater than 2.5 MPa, and the maximum can reach 4 MPa. Hydrogen produced by the electrolysis of water mainly contains impurities such as water vapor and oxygen. If the hydrogen source used by an MHHC is 99.99% pure hydrogen from the coal chemical industry, and the output hydrogen pressure is generally greater than 10 MPa, it is necessary to consider that the content of carbon monoxide ($\leq$5 ppm) can be harmful to hydrogen compression materials. Therefore, the first-stage hydrogen compression material should have a good tolerance to impurity gases such as water vapor and oxygen, and a certain resistance to carbon monoxide. Due to the purifying effect that the first-stage hydrogen compression material has on hydrogen, the poisoning degree of the second-stage, third-stage, and fourth-stage hydrogen compression materials via impurity gases can be reduced.

MHHCs need to be used within a certain temperature range, and using water as a heat exchange medium has many advantages. Compared to oil, water has a larger specific heat capacity, which is conducive to the stable operation of the system. Water is environmentally friendly and can use industrial wastewater and waste heat. Therefore, the best working temperature for hydrogen compression materials is between room temperature (298 $\pm$ 10 K) and 373 K.

All the stages of MHHCs' output pressures should meet the high-pressure hydrogen demand for most scenarios. The first-stage hydrogen compression material, with regard to hydrogen storage and purification, should provide a hydrogen source for the second-stage hydrogen compression material, and its output pressure is about 8 MPa at a high temperature (363 K in this paper). The output pressure of the second-stage hydrogen compression material is greater than 20 MPa at a high temperature, which can fill the long tube trailer and low-pressure gas cylinder with hydrogen. The output pressure of the third-stage hydrogen compression material is greater than 40 MPa at a high temperature, which can fill the 35 MPa high-pressure hydrogen storage tank with hydrogen. The output pressure of the fourth-stage hydrogen compression material at a high temperature is about 80 MPa, which can fill the 70 MPa high-pressure hydrogen storage tank with hydrogen.

Based on the above considerations, the $LaNi_5$ hydrogen storage alloy was selected as the first-stage hydrogen compression material, which has the advantages of a strong toxicity resistance, excellent hydrogenation/dehydrogenation kinetics, and a good cyclic stability [18]. The $TiCr_2$ hydrogen storage alloy was selected as the second- and third-stage hydrogen compression materials [30]. Its advantages include a higher intrinsic plateau pressure, better kinetic performance, and higher hydrogen storage capacity than the $LaNi_5$ alloy. The fourth-stage hydrogen compression material was the $ZrFe_2$-type hydrogen storage alloy, whose advantage is that its inherent plateau pressure can be as high as 1000 MPa [24,25], which makes it easy to develop hydrogen compression materials with a high-pressure hydrogen output.

In order to obtain all the stages of the target hydrogen storage alloys between room temperature and 363 K, the compositions of the alloys were adjusted by the multi-element alloying method, and the desired effect was achieved by alloy preparation technology. The plateau pressure of the binary $LaNi_5$ alloy could be adjusted between 0 and 10 MPa. The replacement of the La by Ce could improve the plateau pressure of the alloy, but the hysteresis coefficient would increase and the hydrogen storage would decrease [22]. The Y element could also increase the alloy's plateau pressure and improve the alloy's kinetic performance. The effect of the hysteresis coefficient was lower than that of the Ce element, but it would increase the alloy's plateau slope [23]. The Ca element could improve the activation property of the alloy, and, as a light hydrogen-absorbing element, could effectively improve the hydrogen absorption capacity and reduce the plateau pressure of the alloy [19,21]. However, the alloy containing the Ca element had a high vapor pressure and was volatile, so it was difficult to control the composition during preparation. Considering the effect of the alloying elements and preparation technology, two $LaNi_5$ hydrogen storage alloys satisfying the MHHC primary hydrogen compression materials were developed:

La$_{0.6}$Ce$_{0.15}$Y$_{0.25}$Ni$_5$ (First-1) and La$_{0.5}$Ce$_{0.2}$Y$_{0.15}$Ca$_{0.15}$Ni$_5$ (First-2). Their compositions and main performance indexes are listed in Tables 1 and 3, respectively. The influence of the alloyed elements on the hydrogen storage properties of the LaNi$_5$ alloys is summarized in Table 4.

**Table 4.** Effect of alloying on hydrogen storage performance of LaNi$_5$ alloys.

| Elements | Capacity | Plateau Pressure | Hysteresis | Slope | Activation Performance |
|----------|----------|------------------|------------|-------|------------------------|
| La | ↑ | ↓ | ↓ | ↓ | ↑ |
| Ce | ↓ | ↑ | ↑ | ↑ | ↓ |
| Y | ↑ | ↑ | ↑ | ↑ | ↑ |
| Ca | ↑ | ↓ | - | - | ↑ |

The plateau pressure of the TiCr$_2$ hydrogen storage alloy could be adjusted between 0 and 20 MPa at room temperature, which mainly solved the problem of a large slope factor and hysteresis coefficient [28]. Mn, Fe, V, and Cu elements were used to adjust the composition of the alloy. Cr and Mn are the two most important transition metal elements on the B side of an AB$_2$ hydrogen storage alloy, and the inter-substitution of Cr and Mn is a common alloy design method. The Cr element is beneficial for reducing an alloy's plateau pressure and hysteresis coefficient, but it makes the hydrogen absorption plateau of the alloy narrow and its hydrogen storage capacity decrease. The Mn element can improve the kinetic properties of an alloy, but significantly increases the alloy's hysteresis coefficient [51]. Due to its small atomic radius, Fe is mainly used to improve the plateau pressure of an alloy [31]. V is a kind of hydrogen absorption element and is an appropriate replacement for the Cr element as it can improve the hydrogen absorption amount of an alloy and reduce its slope, but its price is expensive. The Cu element can increase the plateau pressure of an alloy through the gap size effect, reduce the alloy's slope, and improve the alloy's properties. However, due to the large atomic weight of the Cu and Fe elements, the hydrogen storage capacity of the alloy will be reduced in large quantities. Based on a comprehensive consideration of these alloying elements, the TiCr$_2$ hydrogen storage alloys, TiCr$_{1.1}$Mn$_{0.7}$V$_{0.1}$Cu$_{0.1}$ and TiCr$_{1.4}$Mn$_{0.2}$Fe$_{0.4}$, which met the requirements of the second- and third-stage hydrogen compression materials of MHHCs, were developed, respectively. Their compositions and main performance indexes are listed in Tables 2 and 3, respectively. The influence of these alloyed elements on the hydrogen storage properties of the Ti-Cr-based alloys is summarized in Table 5.

**Table 5.** Effect of alloying on hydrogen storage performance of Ti-Cr-based alloys.

| Elements | Capacity | Plateau Pressure | Hysteresis | Slope | Activation Performance |
|----------|----------|------------------|------------|-------|------------------------|
| Cr | ↓ | ↓ | ↓ | - | - |
| Mn | - | - | ↑ | - | ↑ |
| V | ↑ | - | - | ↓ | - |
| Cu | ↓ | ↑ | - | ↓ | - |
| Fe | ↓ | ↑ | - | - | - |

The hydrogen absorption/desorption pressure of the ZrFe$_2$-type hydrogen storage alloy at room temperature was ~60 MPa, so it was necessary to further reduce the plateau pressure to improve the characteristics of the plateau. The hydrogen-absorbing element Ti was used to replace the partial Zr and the appropriate plateau pressure was achieved by adjusting the Ti/Zr ratio [25]. Replacing part of the Fe element with the V element could effectively reduce the plateau pressure and the hysteresis coefficient, improving the hydrogen storage capacity of the alloy [33]. Comprehensively considering the effect of the alloying elements, the hydrogen storage alloy Zr$_{0.2}$Ti$_{0.8}$Fe$_{1.7}$V$_{0.3}$, a fourth-stage hydrogen compression material satisfying the MHHCs, was developed. Its compositions and main

performance indexes are listed in Tables 2 and 3, respectively. The influence of the alloyed elements on the hydrogen storage properties of the Zr-Fe-based alloys is summarized in Table 6.

**Table 6.** Effect of alloying on hydrogen storage performance of Zr-Fe-based alloys.

| Elements | Capacity | Plateau Pressure | Hysteresis | Slope | Activation Performance |
|---|---|---|---|---|---|
| Ti | ↓ | ↓ | ↓ | ↓ | ↑ |
| V | ↑ | ↓ | ↓ | ↑ | - |

The above 1–4 stage alloys could realize the goals of providing high-pressure pure hydrogen with water as its heat exchange medium and promoting the rapid and good development of a hydrogen energy field through the construction of MHHCs. However, the difficulties in the performance testing of these hydrogen storage alloys with high-pressure hydrogen output characteristics and application efficiency declines caused by non-ideal plateau characteristics are still challenges to be faced (discussed in this paper in Section 2.5). In the future, it is necessary to continue to optimize the comprehensive properties of these alloys, such as their pressure plateaus and hydrogen storage capacities, and to try to improve the accuracy and repeatability of the alloy performance testing.

## 2.2. X-ray Diffraction

Figure 1 shows the XRD finishing patterns of the two first-stage $LaNi_5$ hydrogen storage alloys, second-stage and third-stage $TiCr_2$ hydrogen storage alloys, and fourth-stage $ZrFe_2$ hydrogen storage alloy. It can be seen that the $LaNi_5$ alloy corresponds to a hexagonal $CaCu_5$-type structure (space group P6/mmm); the $TiCr_2$ alloy mainly contains hexagonal $MgZn_2$ phase (C14) and a small amount of $Ti_3Cr_3O$ heterophase. The $ZrFe_2$ alloy is composed of hexagonal $MgZn_2$ phase (C14). Table 7 lists the component phase abundance and lattice constants of the five alloys.

**Table 7.** Lattice parameters and phase abundance of 1–4 stage hydrogen storage alloys.

| Alloys | Phase | Abundance/wt.% | a/Å | c/Å | V/Å3 | Parameters of Fit |
|---|---|---|---|---|---|---|
| **First-1** | $CaCu_5$ | 100% | 4.9596(1) | 3.9885(1) | 84.96(1) | Rw = 4.20<br>Rp = 2.53 |
| **First-2** | $CaCu_5$ | 100% | 4.9610(1) | 3.9862(1) | 84.97(1) | Rw = 4.08<br>Rp = 2.57 |
| **Second** | C14 Laves<br>$Ti_3Cr_3O$ | 91.85<br>8.15 | 4.8700(1)<br>11.3090(7) | 7.9880(2)<br>11.3090(7) | 164.07(1)<br>1446.46(8) | Rw = 5.23<br>Rp = 3.59 |
| **Third** | C14 Laves<br>$Ti_3Cr_3O$ | 93.35<br>6.85 | 4.8619(1)<br>11.2823(2) | 7.97338(1)<br>11.2823(2) | 163.22(1)<br>1436.14(8) | Rw = 4.06<br>Rp = 2.87 |
| **Fourth** | C14 Laves | 100% | 4.8816(1) | 7.9482(2) | 164.03(1) | Rw = 3.57<br>Rp = 2.55 |

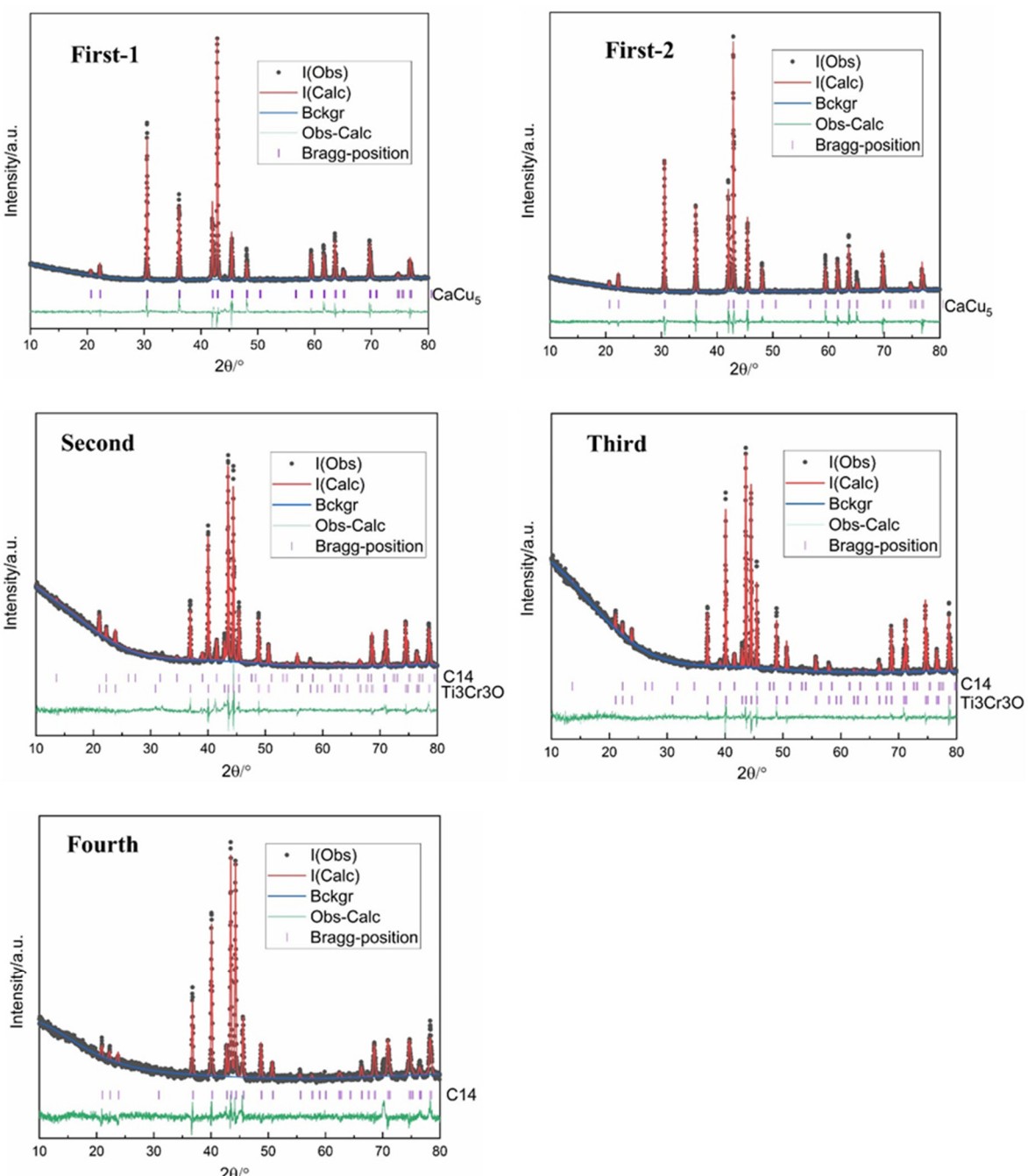

**Figure 1.** Results of the Rietveld analysis for 1–4 stage hydrogen storage alloys.

The lattice constant of the binary $LaNi_5$ hydrogen storage alloy is $a$ = 5.0201 Å, $c$ = 3.9881 Å, and cell volume $V$ = 87.03 Å$^3$ [52]. The atomic radius of the La element is 1.87 Å. The La element in the quaternary $La_{0.6}Ce_{0.15}Y_{0.25}Ni_5$ alloy (First-1) is partially replaced by the Ce (1.82 Å), Y (1.81 Å) elements with smaller atomic radii, thus reducing the lattice constant $a$ and cell volume $V$. The La element in the quinary $La_{0.5}Ce_{0.2}Y_{0.15}Ca_{0.15}Ni_5$ alloy (First-2) is partially replaced by the Ce and Y elements with smaller atomic radii, as well as the Ca (1.96 Å) element with a larger atomic radius. Nonetheless, the lattice constant $a$ and cell volume $V$ also decrease (slightly larger than First-1), which is related to the abnormal decrease in the lattice constant of the cerium-containing alloy that is caused by the change in the valence state of the Ce [53]. The lattice constant $c$ of the $LaNi_5$ alloy is mainly related to the radius of the Ni atom on the B side. Since the Ni in First-1 and First-2 is not replaced by other elements, the $c$ value is similar to that of the binary $LaNi_5$ [52].

The modified $TiCr_2$ alloy is generally still a single-phase C14 structure, and the lattice constant and cell volume vary with the atomic radii of the alloyed elements. The Cr element is usually partially replaced by Mn and Fe. When Mn partially replaces Cr, the $a$ value of the alloy is slightly reduced, but the effects on the $c$ value and $V$ value may be different [31,54]. Fe partially replaces Cr and the $a$, $c$, and $V$ values of the alloy decrease [31,54,55]. Therefore, the cell volume of the second-stage alloy, $TiCr_{1.1}Mn_{0.7}V_{0.1}Cu_{0.1}$, and the third-stage alloy, $TiCr_{1.4}Mn_{0.2}Fe_{0.4}$, is similar in this paper. The $Ti_3Cr_3O$ heterophase in the alloys may be related to the high melting power (20~30 KW) and the short cooling time (8~10 min) during the alloy preparation (see Section 3).

The binary $ZrFe_2$ hydrogen storage alloy has a cubic C15 Laves phase structure [33]. In this paper, the fourth-stage $Zr_{0.2}Ti_{0.8}Fe_{1.7}V_{0.3}$ alloy transforms into a hexagonal C14 Laves phase structure because the Ti partially replaces the Zr and the V partially replaces the Fe. The atomic radius of the Ti (1.45 Å) is smaller than that of the Zr (1.60 Å), and the atomic radius of the V (1.31 Å) is larger than that of the Fe (1.24 Å). Compared to the lattice constants ($a$ = 5.023 Å; $b$ = 8.205 Å; and $V$ = 179.35 Å$^3$) of the $Zr_{1.05}Fe_{1.8}Mo_{0.2}$ (Mo has an atomic radius of 1.36 Å) alloy with a C14 Laves phase structure [33], the $a$ (4.8816 Å), $c$ (7.9482 Å), and $V$ (164.03 Å$^3$) values of the $Zr_{0.2}Ti_{0.8}Fe_{1.7}V_{0.3}$ alloy are significantly reduced. The results show that the lattice constants of the $Zr_{0.2}Ti_{0.8}Fe_{1.7}V_{0.3}$ alloy are significantly affected by the substitution of the Zr by Ti elements.

### 2.3. PCT Test Analysis

The reaction of the hydrogen storage alloy (M) with hydrogen ($H_2$) to form metal hydride ($MH_x$) is a reversible heat ($Q$)-driven process:

$$M + x/2 \, H_2 \leftrightarrows MH_x + Q \tag{3}$$

The equilibrium characteristic of Reaction (3) is the relationship between the hydrogen pressure ($P$), hydrogen concentration in the solid phase ($C$), and temperature ($T$) (PCT diagram), and determines the thermodynamic properties of the hydrogenation of a particular hydrogen storage alloy. At a certain temperature, the hydrogen concentration $C$ absorbed in M exceeds the concentration of the saturated solid solution ($\alpha$ phase) to form a hydride phase ($\beta$ phase), which shows the characteristics of the first-order phase transition. With the increase in the $C$, the hydrogen pressure $PH_2$ remains unchanged, which is called plateau pressure, and the plateau width corresponds to the reversible hydrogen storage capacity of the alloy. The equilibrium of Reaction (3) in the plateau region can be described by the Van't Hoff equation:

$$\ln PH_2 = -\Delta S^0/R + \Delta H^0/RT \tag{4}$$

$\Delta S^0$ and $\Delta H^0$ are the standard formation entropy and standard formation enthalpy of the hydrides, respectively, and R is the gas constant [1].

Using a PCT tester to measure the PCT curves for at least three different temperatures, the hydrogenation/dehydrogenation plateau pressure ($P_a$ and $P_d$) and maximum hydrogen storage capacity at a certain temperature can be obtained. Equation (2) was used to calculate the plateau slope factor $S_f$ and Equation (1) for the plateau hysteresis coefficient $H_f$. According to Equation (4), Van't Hoff graphs (ln $PH_2$-1/T graphs) were drawn and the $\Delta S^0$ and $\Delta H^0$ were calculated. For the high-pressure alloy, the hydrogen emission pressure at a high temperature $T_H$ ($T_H$ is 363 K in this paper) can be obtained by an extrinsic method.

The multi-stage compression operation puts forward higher requirements for regulating the PCT characteristics of the hydrogen compression materials. Figure 2 shows the PCT curves of the 1–4 stage alloys at different temperatures, among which, the PCT curves of the first-stage alloys (First-1 and First-2) at a high temperature $T_H$ (363 K) can be directly measured. The highest temperature of the second-, third-, and fourth-stage alloys is 353 K due to the limitation of the testing capability of the instrument. As can be seen from Figure 2, the pressure plateau of the first-stage alloy is relatively flat but has a large

hysteresis, while the pressure plateau of the second-, third-, and fourth-stage alloys is steep but has a small hysteresis. The PCT characteristic parameters of all the alloys are listed in Table 3. Figure 3 shows the Van't Hoff diagram of the 1–4 stage hydrogen storage alloys. The inclination and hysteresis of the pressure plateau are non-idealized properties of the hydrogen storage alloy, and it is difficult to reach the ideal state at the same time.

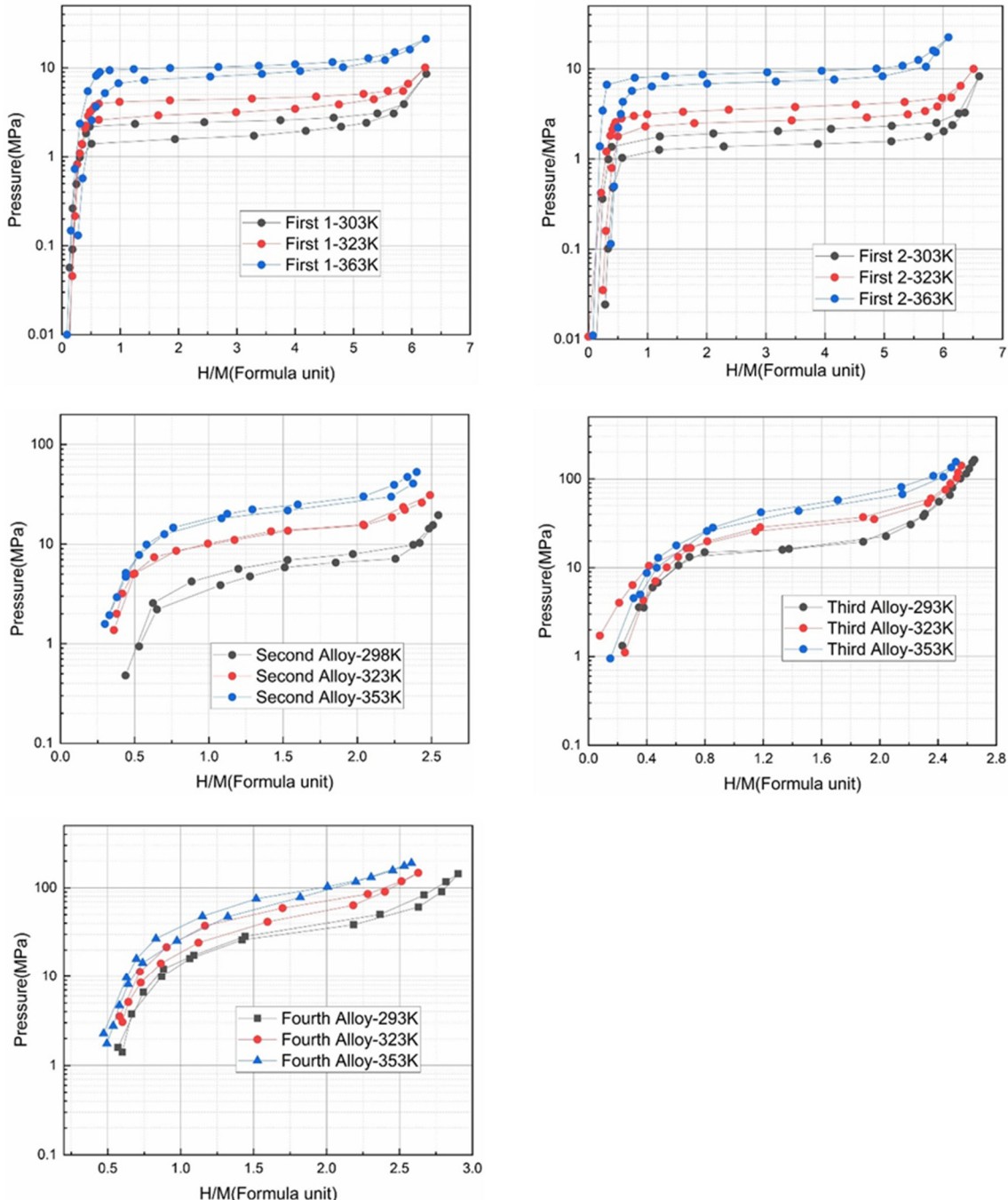

**Figure 2.** PCT curves of 1–4 stage hydrogen storage alloys at different temperatures.

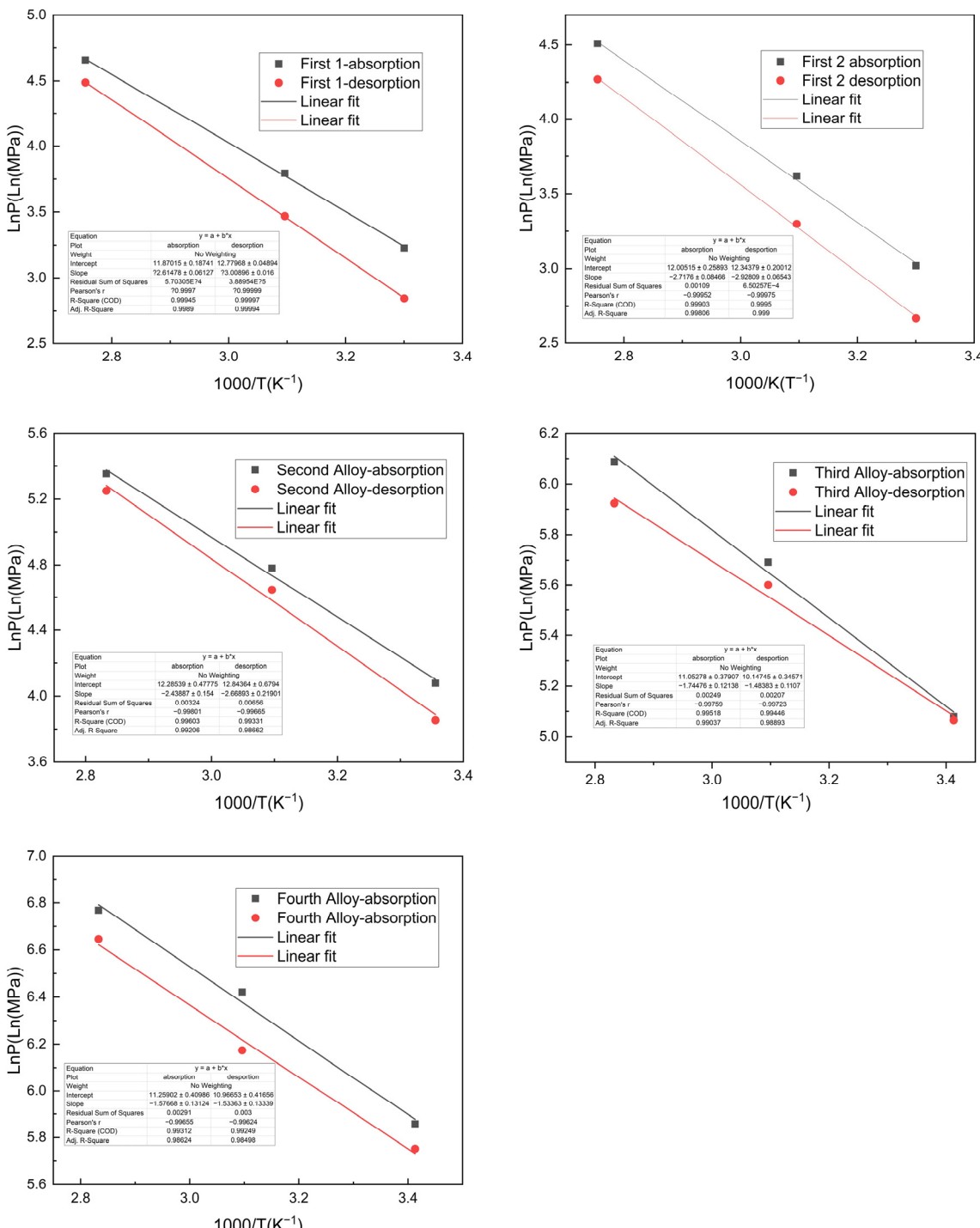

**Figure 3.** Van't Hoff diagram of 1–4 stage hydrogen storage alloys.

The compression ratios $R_P$ [$=P_d$ $(T_H)/P_a$ $(T_L)$] of the 1–4 stage alloys are 3.53 (First-1) or 3.49 (First-2), 4.24 (second), 2.67 (third), and 2.42 (fourth), respectively. The total compression ratios of the constructed MHHC system are 96.71 or 95.61 (the product of the compression ratios at all the stages) [5]. The compression ratio mainly reflects the compression capacity of the MHHCs. The output pressures of the alloys in this paper are relatively large and, due to this and the plateau hysteresis, the compression ratio is at a moderate level and needs to be further improved.

In the MHHC system, which is constructed of hydrogen compression material, each stage of the hydrogen compression material absorbs hydrogen at a low temperature $T_L$

(usually room temperature) with a maximum hydrogen absorption capacity of $C_A$ (corresponding to the right end of the pressure plateau in the PCT curve), and releases hydrogen at a high temperature $T_H$ (363 K) with a maximum hydrogen emission of $C_B$ (corresponding to the left end of the pressure plateau in the PCT curve). Then, the cycle yield of the MHHC system can be expressed as the effective hydrogen compression capacity, $\Delta C$:

$$\Delta C = C_A(T_L) - C_B(T_H) \qquad (5)$$

Based on this calculation, the $\Delta C$s of the First-1 and First-2 alloys are about 5.25 H/f.u. (1.01 wt%) and 5.90 H/f.u. (1.42 wt%), respectively. The highest temperature measured by the PCT curves of the second-, third-, and fourth stage alloys is 353 K. Calculated according to the amount of hydrogen released at 353 K, the $\Delta C$s of the second-, third-, and fourth stage alloys are about 1.85 H/f.u. (1.18 wt%), 1.85 H/f.u. (1.19 wt%), and 1.90 H/f.u. (1.13 wt%), respectively. The $\Delta C$ values of the second-, third-, and fourth stage alloys at 363 K should be slightly lower than those calculated above due to the temperature increasing, the plateau pressure increasing, the plateau narrowing, and the $C_B$ point shifting to the right and increasing. Considering the compression ratios and cycle yields of the hydrogen compression materials, the First-2 alloy is superior to the First-1 alloy as a primary hydrogen compression material. The cyclic yield reflects the hydrogen supply capacity of MHHCs. The effective hydrogen storage capacity of an alloy can be increased by adjusting the alloy's composition and improving the characteristics of its plateau.

The plateau pressure of the hydrogen storage alloys is related to the types, phase compositions, and cell parameters of the alloys. The two first-stage alloys in this paper are both $CaCu_5$-type single-phase structures, their lattice constants and cell volumes are very similar, and their plateau pressures are also similar. The main phase of the second-, third-, and fourth stage alloys is the C14 Laves phase, and their lattice constants and cell volumes are also very similar, but their plateau pressures vary greatly, which may be related to the composition elements of the alloys, and needs further study.

### 2.4. Cyclic Performance Test

MHHCs are a continuous cycle system of the hydrogen absorption and desorption of hydrogen compression materials. The toxic effect of some of the impurities in the hydrogen on hydrogen compression materials seriously affects the cyclic performance of the materials. In order to investigate this effect, the cyclic performances of the first- and second-stage alloys were tested using ordinary pure hydrogen. Table 8 lists the gas composition test results, which meet Chinese standards. Due to the purification effect of the first- and second-stage alloys on the hydrogen, the content of the impurity gases in the hydrogen entering the third- and fourth-stage alloys will be significantly reduced. Moreover, the third- and fourth-stage alloys are of the same type as the second-stage alloy, so the cyclic performances of the third- and fourth-stage alloys in ordinary pure hydrogen were not investigated.

**Table 8.** Composition of ordinary pure hydrogen.

| Gas Composition | Standard Indicators | Measured Content |
|---|---|---|
| Hydrogen ($H_2$) (%) | $\geq 99.99$ | 99.99 |
| Oxygen ($O_2$) (ppm) | $\leq 5$ | 3 |
| Nitrogen ($N_2$) (ppm) | $\leq 60$ | 50 |
| Carbon monoxide (CO) (ppm) | $\leq 5$ | 3 |
| Carbon dioxide ($CO_2$) (ppm) | $\leq 5$ | 2 |
| Methane ($CH_4$) (ppm) | $\leq 10$ | 5 |
| Water vapor ($H_2O$) (ppm) | $\leq 10$ | 8 |

Figure 4 shows the change curves of the hydrogen storage capacities of the first-stage alloys (First-1 and First-2) and second-stage alloy for 10 cycles, respectively. It can be seen

that the hydrogen storage capacity retention rates are 99.0%, 97.7%, and 98.6%, respectively, which have a good cycle stability. The cycle stability of the First-2 alloy is less than that of the First-1 alloy due to the inclusion of the Ca element.

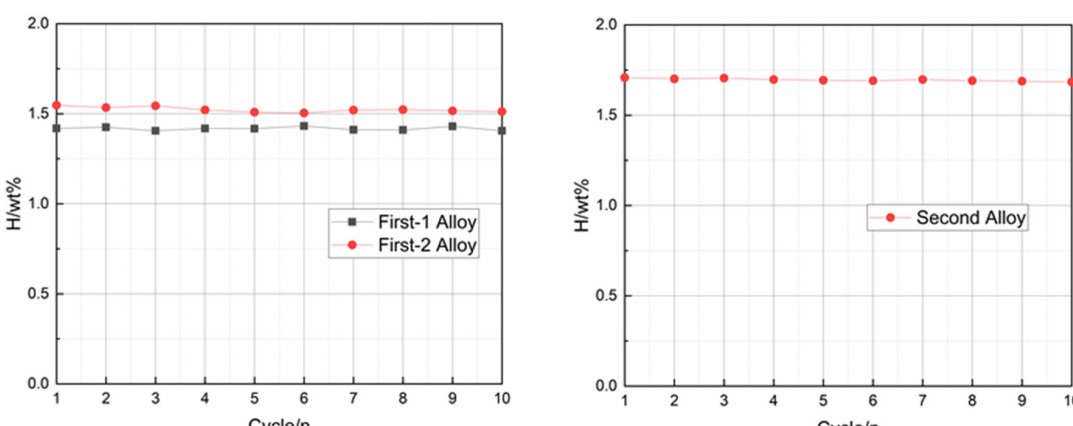

**Figure 4.** Hydrogenation/dehydrogenation cycle curves of 1–2 stage alloys in ordinary pure hydrogen.

*2.5. Application Analysis*

There are two problems to be considered in the practical application of hydrogen storage alloys as hydrogen compression materials. First is the question of: is the plateau pressure of the hydrogen storage alloy consistent with the pressure designed in the MHHC system? The second question is: can the former and latter hydrogen storage alloys work together efficiently?

The hydrogen compression materials used in MHHCs are related to the reaction characteristics at a high temperature and high pressure. It is difficult to measure the high-pressure characteristics of alloys with the current PCT test device, such as the second-, third-, and fourth stage hydrogen storage alloys in this paper. In order to solve this problem, we can use the Van't Hoff equation to draw the linear relationship between the pressure ($\ln P$) and temperature reciprocal ($1/T$) (see Figure 3) and obtain the alloy pressure under a certain high-temperature condition by linear extrapolation. However, due to factors such as hydrogen fugacity and temperature fluctuation, the Van't Hoff diagram deviates from the linear relationship at a high temperature [17,31], and the corrected pressure value is lower than the pressure value obtained by extrapolation [56]. Therefore, the high-temperature output pressure of the hydrogen storage alloy obtained by the Van't Hoff diagram should be higher than the target output pressure, and its applicability should be further evaluated through practical application.

During the actual operation of an MHHC system, in order to ensure the matching and coupling between the alloys at all the stages, the hydrogenation pressure of the latter alloy at a low temperature must be lower than that of the former alloy at a high temperature. In an ideal state, the matching coupling between two adjacent alloys is considered by using the plateau pressure mid-value of the PCT curve (as shown in Figure 5), that is, the plateau pressure mid-value of the former alloy at a high temperature is greater than that of the latter alloy at a low temperature. However, the pressure plateau of any alloy is inclined, for example, the second-, third-, and fourth stage alloys in this paper have obvious plateau slopes. In this case, the dehydrogenation process can be completed only when the former-stage alloy dehydrogenates to the low point of the plateau pressure (left side of the plateau), and the hydrogen absorption process can be completed only when the latter-stage alloy absorbs hydrogen to the high point of the plateau pressure (right side of the plateau) (as shown in Figure 6). Comparing Figures 5 and 6, it can be seen that the plateau pressures of the second-, third-, and fourth stage alloys in this paper cannot fully meet the practical application. The minimum pressure of the former-stage alloys for high-temperature dehydrogenation needs to be increased, or the maximum hydrogen absorption pressure of the latter-stage alloys at a low temperature needs to be reduced.

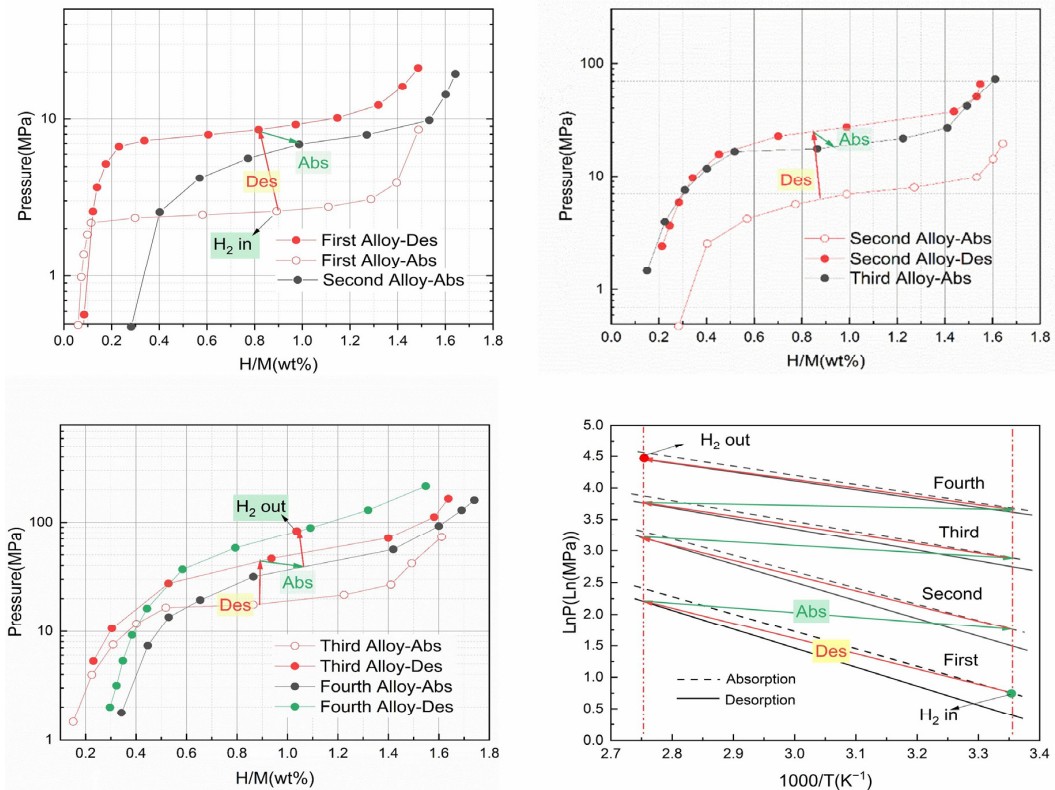

**Figure 5.** Matching coupling relationship of 1–4 stage hydrogen storage alloys in ideal state.

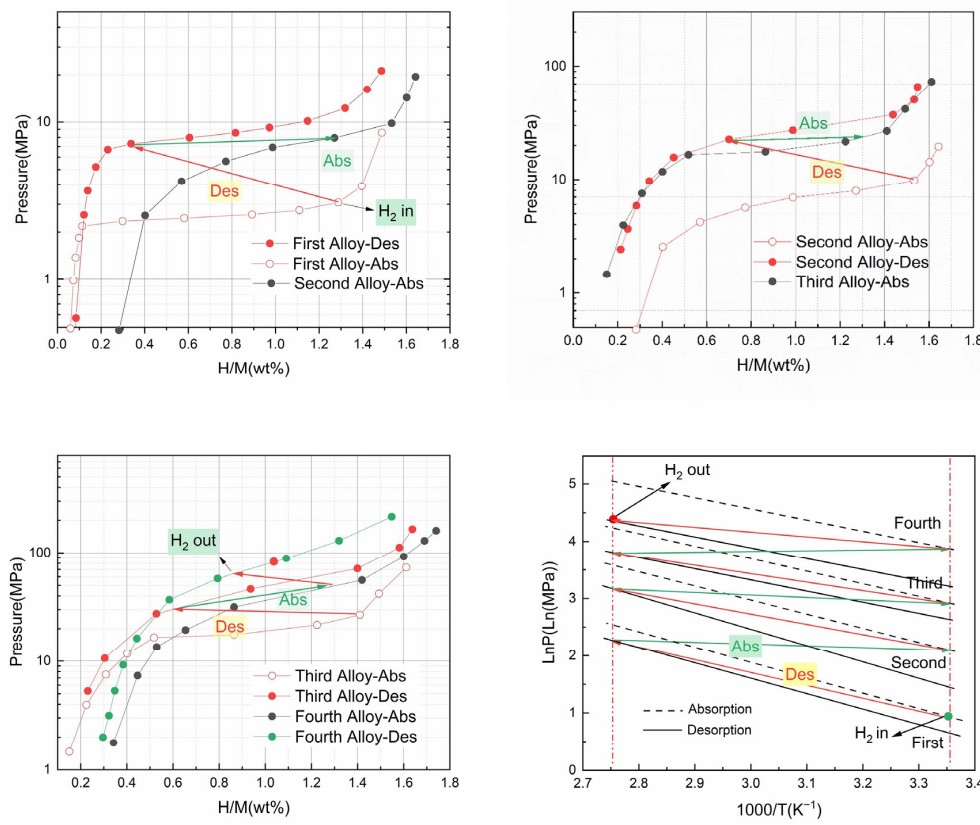

**Figure 6.** Matching and coupling relationship of 1–4 stage hydrogen storage alloys in actual state.

## 3. Materials and Methods

The metal raw materials required for the alloy preparation were calculated and weighed according to the composition formula. The purity of the metal raw materials was ≥99 wt%. Considering the burning loss of each single metal in the melting process, the La, Ce, and Y in the first-stage alloy were added by 1 wt%, and the Ca was added by 20 wt%. The Ti, Zr, Mn, and V in the second-, third-, and fourth-stage alloys were added with 3 wt%, 2 wt%, 5 wt%, and 6 wt%, respectively.

The first-stage alloy was prepared in a vacuum induction melting furnace under the protection of a 0.055 MPa argon atmosphere. The smelting process was: preheating a power of 7 KW for 3–4 min, maintaining a power of 15 KW until the alloy was completely melted, holding a power of 12 KW for 3–4 min, and then pouring into a water-cooled ingot mold with a sample weight of 1.5 kg each time.

The second-, third-, and fourth-stage alloys were melted in a magnetic levitation induction furnace under the protection of a 0.055 MPa argon atmosphere. The mass of each sample was 80 g. The smelting process was: 10–12 KW for 1 min during preheating, 20 KW for melting, 25–30 KW for 1 min so that the alloy was completely alloyed, and finally reducing the power to 0, then cooling for 8–10 min, after which, the sample was taken out. In order to ensure a uniform alloy composition and structure, each alloy sample was melted and turned over three times.

After the sample was taken out, the oxide layer on the surface was removed, and after crushing, it was passed through a 200 mesh screen. The +200 mesh powder was used for the composition test and PCT performance test. The −200 mesh powder was used for the X-ray powder diffraction test.

The composition of the alloy sample was tested by a Thermo Electron-iCAP 6300 Inductively Coupled Plasma Emission Spectrometer (ICP), and the actual composition contents of the various elements in the alloy sample were calculated according to the results.

The crystal structure of the alloy was determined by a Philips-PW 1700X powder diffractometer (XRD). The test conditions were: a Cu target, a K$\alpha$ radial, a tube voltage of 40 kV, a tube current of 40 mA, a scanning range of 10°–80°, and a scanning speed of 0 01°/s. The phase structure analysis software used was Jade 6.0 and the Rietveld full-spectrum fitting and structure refinement used GSAS software [57] to obtain the phase abundance and crystal cell parameters.

The PCT performances of the hydrogen storage alloys were tested with a Sievert-type device (manufactured by the Beijing Research Institute of Engineering Technology). The PCT performances of the alloys for the third and fourth stages were tested in MSU in a high-pressure device, as described earlier in [58]. During the activation process, the +200 mesh sample ~2 g was taken and put into the sample tank. The sample tank was evacuated for 0.5 h at 423 K to ensure that the vacuum was less than $5 \times 10^{-4}$ MPa. The sample was cooled to 313 K in a water bath and charged with a certain hydrogen pressure, according to the design characteristics of the alloy for the sample activation treatment. The high-temperature vacuum low-temperature hydrogen charging process was repeated to ensure that the sample was fully activated, and then the PCT curve test was carried out at different temperatures. Before testing the PCT curve, the sample tank should be evacuated at 423 K for 0.5 h to ensure the complete dehydrogenation of the sample. In order to ensure the complete balance of each measuring point during the PCT performance test, the time of each measuring point should be maintained for more than 30 min until the pressure change of ≤0.001 MPa/min. The purity of the hydrogen used in the activation and PCT curve tests was 99.999%, and the purity of the hydrogen used in the normal hydrogen cycle performance test was 99.99%.

## 4. Conclusions

According to the demand of metal hydride–hydrogen compressors (MHHC) for hydrogen compression materials, the first-stage $LaNi_5$, the second- and the third-stage $TiCr_2$, and the fourth-stage $ZrFe_2$ hydrogen storage alloys were developed. The output pressures

of the 1–4 stage hydrogen storage alloys at 363 K were 8.90 MPa, 25.04 MPa, 42.97 MPa, and 84.73 MPa, respectively. The maximum output hydrogen pressure exceeded 80 MPa, which could meet the requirements of the 20 MPa, 35 MPa, and 70 MPa high-pressure hydrogen injections.

The LaNi$_5$ type alloy was a single-phase CaCu$_5$ type hexagonal structure, the TiCr$_2$-type alloy main phase was a hexagonal C14 Laves phase and Ti$_3$Cr$_3$O heterophase, and the ZrFe$_2$ type alloy was composed of a hexagonal C14 Laves single phase. The total compression ratio of the MHHC system constructed with the 1–4 stage alloys exceeded 95, and the hydrogen yield of each stage was more than 1 wt%. The retention rate of the hydrogen storage capacities of the first- and second-stage alloys in the 99.99 wt% ordinary pure hydrogen containing H$_2$O (8 ppm), O$_2$ (3 ppm), and CO (3 ppm), for 10 cycles of hydrogenation/dehydrogenation, exceeded 98%. The deviation between the plateau pressure and the system design output pressure, as well as the efficient and cooperative coupling of the adjacent two stage alloys, should also be considered in the practical application of the four-stage hydrogen storage alloys.

**Author Contributions:** S.M., E.M. and M.P. completed the preliminary statistical model design of alloy composition and the high pressure PCT tests of the third and fourth stage alloys. H.-Z.Y., X.Z., Y.-Y.Z., B.-Q.L., J.X. and W.X. completed the application design of alloy composition, alloy preparation and all other tests. All authors have read and agreed to the published version of the manuscript.

**Funding:** This research was funded by the National Key Research and Development Program of China (2018YFE0124400) and Major Science and Technology Projects of Inner Mongolia (2021ZD0029). Cooperative project BRICS2019-032 was supported by BRICS national annexes of Russian Federation, China and India. Part of this work was performed according to the Development program of the Interdisciplinary Scientific and Educational School of Lomonosov Moscow State University "The future of the planet and global environmental change", and was supported by the Ministry of Science and Higher Education of the Russian Federation, projects # AAAA-A16-116053110012-5 and 122012400186-9.

**Data Availability Statement:** Data is contained within the article.

**Conflicts of Interest:** The authors declare no conflict of interest.

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
