# Peer review of "Hydrogen Compression Materials with Output Hydrogen Pressure in a Wide Range of Pressures Using a Low-Potential Heat-Transfer Agent"

_inorganics, doi:10.3390/inorganics11050180_

Round 1

Reviewer 1 Report

The submitted manuscript discusses the design of 4-stage MHHC materials and some practical aspects of operating MHHC in a real condition. Some questions and comments are as follows:

 1) In Table 1, it is recommended to clearly mark the calculated Pa and Pd values and to explain in the caption. It is confusing to see the­ tabulated Pa and Pd values at the temperatures with no PCT data in Figure 2.

 2) In page 8, please clarify what pressure values are used to get CA(TL) and CB(TH) in Eq. 5. In the following explanation, the authors wrote that deltaC values at 363K should be slightly lower because of decrease of dehydrogenation at increasing temperature. I guess the opposite is true. Since CB(TH) is smaller at increasing temperature under fixed pressure, deltaC at 363K should be slightly higher unless the pressure condition changes.

 3) It is recommended to revise the abstract to reflect the conclusion that the combination of the four alloys cannot fulfill 80 MPa compression in a real application.

Reviewer 2 Report

The article describes the development of four hydrogen storage alloys for use as hydrogen compression materials in metal hydride-hydrogen compressors (MHHCs) for high-pressure hydrogen filling in the hydrogen energy field. The statistical model and research experience were used to determine the preliminary composition of the hydrogen storage alloys. The alloys were prepared by high-temperature melting, and their composition, structure, and hydrogenation/dehydrogenation plateau characteristics were tested by various methods. The output pressure of the four-stage hydrogen storage alloys at 363K was found to be 8.90 MPa, 25.04 MPa, 42.97 MPa, and 84.73 MPa, respectively, meeting the requirements of high-pressure hydrogen injection for MHHC. Overall, the article is well-written and presents a clear overview of the development of the four hydrogen storage alloys. However, there are some specific areas that could be improved:

1.      The introduction could be more comprehensive, providing a background on MHHCs and their importance in the hydrogen energy field. The author should also provide more details on the current state-of-the-art in hydrogen compression materials and how the proposed alloys improve upon existing materials. 

2.      The methods section could be more detailed, providing more information on the specific conditions used for the high-temperature melting and hydrogenation/dehydrogenation testing.

3.      The results section could be presented more clearly. The author should consider providing tables or figures to better illustrate the data and highlight the key findings.

4.      The discussion section could be expanded to provide more insights into the significance of the results and their potential impact in the field. The author should also discuss any limitations or challenges associated with the proposed alloys and suggest possible future directions for research.

5.      The language and grammar need to be improved in some places. The author should pay attention to sentence structure, word choice, and overall clarity of expression to ensure that the paper is easily understood by the reader.

Overall, the article provides valuable insights into the development of hydrogen storage alloys for use as hydrogen compression materials in MHHCs. However, the author should consider revising and expanding some sections to provide a more comprehensive and detailed analysis of the research findings.

Round 2

Reviewer 2 Report

Accept